# DriveGPT4: Interpretable End-to-end Autonomous Driving via Large Language Model

## Abstract

In the past decade, autonomous driving has experienced rapid development in both academia and industry. However, its limited interpretability remains a significant unsolved problem, severely hindering autonomous vehicle commercialization and further development. Previous approaches utilizing small language models have failed to address this issue due to their lack of flexibility, generalization ability, and robustness. Recently, multimodal large language models (LLMs) have gained considerable attention from the research community for their capability to process and reason non-text data (e.g., images and videos) by text. In this paper, we present DriveGPT4, an interpretable end-to-end autonomous driving system utilizing LLMs. DriveGPT4 is capable of interpreting vehicle actions and providing corresponding reasoning, as well as answering diverse questions posed by human users for enhanced interaction. Additionally, DriveGPT4 predicts vehicle low-level control signals in an end-to-end fashion. These capabilities stem from a customized visual instruction tuning dataset specifically designed for autonomous driving. To the best of our knowledge, DriveGPT4 is the first work leveraging LLMs for interpretable end-to-end autonomous driving. When evaluated on multiple tasks alongside conventional methods and video understanding LLMs, DriveGPT4 demonstrates superior qualitative and quantitative performance.

## 1 Introduction

Over the past decade, there has been remarkable growth in the field of autonomous driving, encompassing both academia and industry (Singh & Saini, 2021; Liu et al., 2021; Parekh et al., 2022). Commercialized autonomous driving systems have been successfully implemented in everyday scenarios, such as harbors, warehouses and urban areas. Commonly, the autonomous vehicle adopts modular designs, including perception, planning, and control. In conventional autonomous driving systems, these modules are implemented by detailed rule-based methods to handle various scenarios. But such a system may fail when unseen cases are met, such as rare accidents.

To ensure that vehicles can effectively handle diverse situations using intelligent actions, data-driven learning-based methods have become a widespread component of modern autonomous driving systems (Zhao et al., 2017; Xue et al., 2019; Xu et al., 2022; 2023a;b). To better integrate and optimize the entire system, some approaches propose training the network in an end-to-end manner, eliminating the need for discontinuous intermediate steps (Prakash et al., 2021; Hu et al., 2023; Chen et al., 2023). By using vehicle-mounted sensor data as input, the end-to-end autonomous driving system can directly predict planned paths and/or low-level vehicle controls. Nonetheless, the end-to-end learning-based autonomous driving system functions as a black box, signifying that humans cannot interpret or comprehend the generated decisions, leading to significant ethical and legal concerns.

In recent years, explainable autonomous driving (Deruyttere et al., 2019; Kim et al., 2019; Atakishiyev et al., 2021; Jin et al., 2023; Malla et al., 2023) has garnered increasing interest due to its potential to demystify the black box. These studies develop large-scale datasets comprising autonomous vehicle data along with language pairs. Language models, such as BERT (Devlin et al., 2018) and GPT (Radford et al., 2018), are trained on these datasets to generate natural language based on input from vehicle-mounted sensor data. However, the capabilities of small language mod-

els are limited, causing most of these systems to produce rigid responses to predefined questions. When faced with new or unexpected inquiries, these methods struggle to deliver satisfactory results.

With the advent of large language models (LLMs), such as ChatGPT (OpneAI, 2023) and LLaMA (Touvron et al., 2023a), end-to-end autonomous driving systems could benefit from improved explanations, given that LLMs possess extensive general knowledge about the world. Moreover, LLMs have the potential to directly generate low-level vehicle controls due to their inherent reasoning capabilities. To achieve this, LLMs need to comprehend videos and understand low-level vehicle controls. Multimodal LLMs have been attracting increasing interest from various research communities, such as computer vision (Li et al., 2022b;a), embodied AI (Driess et al., 2023; Liang et al., 2023), and biomedicine (Karabacak & Margetis, 2023; Li et al., 2023a). These studies propose to project multimodal input from image, audio, video, control, and other spaces into the text domain, allowing LLMs to understand and process this multimodal data as text. To the best of our knowledge, no existing paper leverages LLMs for interpretable end-to-end autonomous driving purposes.

In this paper, we introduce DriveGPT4, an interpretable end-to-end autonomous driving system that utilizes large language models. "4" represents multimodality inspired by MiniGPT4 (Zhu et al., 2023). DriveGPT4 takes as input a video sequence captured by a front-view RGB camera, and then predicts the control signal for the next step (i.e., vehicle speed and turning angle). At the same time, human users can converse with DriveGPT4, which can provide natural language responses, such as describing the vehicle's actions and explaining the reasoning behind its behavior. To train DriveGPT4 to communicate like a human, we follow LLaVA (Liu et al., 2023) and create a visual instruction tuning dataset based on the BDD-X dataset (Kim et al., 2018) using ChatGPT. The contributions of this paper are summarized as follows:

- We develop a new visual instruction tuning dataset for interpretable autonomous driving.

- We present a novel multimodal LLM called DriveGPT4, based on Valley (Luo et al., 2023). Fine-tuned on the created dataset, DriveGPT4 can process multimodal input data and provide text responses as well as predicted control signals.

- We evaluate all methods on multiple tasks, and DriveGPT4 outperforms all baselines. In addition, DriveGPT4 can handle unseen scenarios with zero-shot generalization to some extent.

## 2 RELATED WORKS

**End-to-end Autonomous Driving.** End-to-end autonomous driving aims to directly predict the vehicle path and low-level control signals based on visual inputs (Bojarski et al., 2016; Xiao et al., 2020; Prakash et al., 2021; Hu et al., 2023; Chen et al., 2023). (He et al., 2016) is considered the first deep learning end-to-end self-driving work. In this study, the authors train a convolutional neural network to control vehicles using monocular images as input. Recent works integrate all system modules by tokenizing module outputs (Hu et al., 2023; Chen et al., 2023), achieving a more powerful and robust control effect. However, these works lack interpretability, which limits their trustworthiness and commercialization potential.

**Interpretable Autonomous Driving.** To address the black box issue in learning-based autonomous driving, some studies employ visualizations (Kim & Canny, 2017; Wang et al., 2021; Saha et al., 2022). However, visual maps can be challenging for non-expert passengers to comprehend. Alternatively, other research utilizes language models to describe vehicle situations with natural languages, such as vehicle actions (Deruyttere et al., 2019; Kim et al., 2019; Jin et al., 2023), vehicle action reasoning (Jin et al., 2023), surrounding object statements (Malla et al., 2023), and discussions of potential risks to the ego vehicle (Malla et al., 2023). Constrained by the limited capacity of smaller language models, these methods can only address predefined human questions and provide inflexible answers, hindering their widespread application in real-world scenarios.

**Multimodal LLM.** Building on the powerful pretrained LLM weights, such as PaLM (Chowdhery et al., 2022; Driess et al., 2023), LLaMA (Touvron et al., 2023a;b), and Vicuna (Peng et al., 2023), multimodal LLMs aim to handle multiple types of input beyond text. Blip (Li et al., 2022a; 2023b) leverages Q-formers to project multimodal input into the text space, while others (Li et al., 2023a; Luo et al., 2023) simply train a fully connected layer as the projector. Multimodal LLMs have been

widely applied to various tasks, such as image understanding (Li et al., 2023b; Liu et al., 2023), video understanding (Luo et al., 2023; Zhang et al., 2023; Wang et al., 2023; Zhu et al., 2023; Li et al., 2023c), medical diagnosis (Li et al., 2023a; Karabacak & Margetis, 2023), and embodied AI (Chowdhery et al., 2022; Driess et al., 2023; Brohan et al., 2023; Liang et al., 2023), etc. Our task is closely related to video understanding and embodied AI. DriveGPT4 is inspired by the former to understand input video data and the latter to predict control signals. Among these works, only a few focus on autonomous driving-related tasks (Fu et al., 2023; Wu et al., 2023; Contributors, 2023). DriveLikeHuman (Fu et al., 2023) can only handle simple simulation scenes, limiting its real-world applicability. NuPrompt (Wu et al., 2023) focuses on object tracking for vehicle perception but does not consider end-to-end driving or vehicle action reasoning. DriveLM (Contributors, 2023) is the most similar work to ours. However, it can only predict high-level plans (e.g., go straight, turn left), which is insufficient for our task. To the best of our knowledge, no existing paper shares our exact scope: leveraging LLMs for interpretable end-to-end autonomous driving.

# 3 DATA GENERATION

**BDD-X Dataset.** Videos and labels are collected from the BDD-X dataset (Kim et al., 2018), which contains approximately 20,000 samples, including 16,803 clips for training and 2,123 for testing. Each clip is sampled into 8 images. Additionally, it provides control signal data for each frame (e.g., vehicle speed and vehicle turning angle). BDD-X offers text annotations about vehicle action descriptions and action justifications for each video clip, as illustrated in Fig. 1.

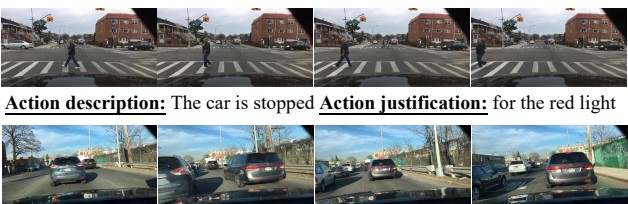

**Action description:** The car is stopped **Action justification:** for the red light

**Action description:** The car pulls into the right lane
**Action justification:** because traffic is moving faster in that lane.

Figure 1: Example of BDD-X labeled data.

**Fixed question-answering.** BDD-X provides three types of labels: vehicle action descriptions, action justifications, and control signal for each video clip. To train the LLM, question-answering (QA) pairs are required. We need to generate a set of questions and use corresponding BDD-X labels as the answer. For example, for a vehicle action description, a question equivalent to "What is the current action of this vehicle?" should be sent to the LLM as the input question. Then, the LLM should generate the response, whose ground truth label is the vehicle action description. Considering there are three types of labels in BDD-X dataset, we create three question sets: $Q_a$, $Q_j$, and $Q_c$. To prevent the LLM from only answering fixed question patterns, each question set should contain multiple expressions of one question.

- $Q_a$ contains questions equivalent to "What is the current action of this vehicle?". A randomly selected question $q_a \in Q_a$ forms a QA pair with the action description label.

- $Q_j$ contains questions equivalent to "Why does this vehicle behave in this way?". A randomly selected question $q_j \in Q_j$ forms a QA pair with the action justification label.

- $Q_c$ contains questions equivalent to "Predict the speed and turning angle of the vehicle in the next frame.". A randomly selected question $q_c \in Q_c$ forms a QA pair with the control signal label.

LLMs can learn to predict and interpret vehicle actions simultaneously. However, these QA pairs have fixed and rigid contents. Due to the lack of diversity, training solely on these QAs will degrade the reasoning ability of LLMs and render them incapable of answering questions in other formats.

**Conversations generated by ChatGPT.** In previous works, ADAPT (Jin et al., 2023) trains a caption network to predict descriptions and justifications. However, the provided description and justification labels are fixed and rigid. If human users wish to learn more about the vehicle and ask everyday questions, past works may fall short. Thus, BDD-X alone is insufficient for meeting the requirements of interpretable autonomous driving. Instruction tuning data generated by Chat-GPT/GPT4 has been proven effective for performance enhancement in natural language processing (Peng et al., 2023), image understanding (Liu et al., 2023), and video understanding (Li et al., 2023c; Zhang et al., 2023). ChatGPT/GPT4 can access privileged information (e.g., image-labeled captions,

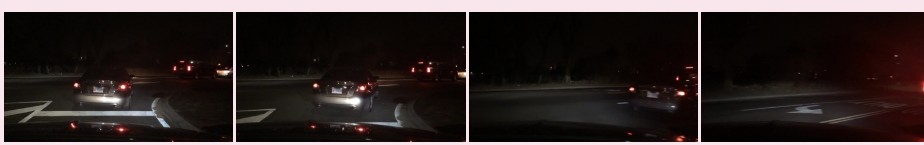

**Caption1: Action description**
The car turns right
**Caption 2: Action justification**
As the road is clear to turn.
**Control signals**
Speed(m/s): 3.91, 3.1, 2.35, 2.92, 3.51, 4.24, 4.85, 5.22
Turning angle(degree): 0.0, -3.08, -5.98, -4.66, -2.91, 1.97, 7.02, 14.46
**Object boxes**
Frame 0: car:[0.298, 0.408, 0.572, 0.756], car:[0.924, 0.408, 1.0, 0.51], car:[0.005, 0.83, 0.995, 0.982], car:[0.737, 0.373, 0.933, 0.522], car:[0.737, 0.373, 0.933, 0.522]
Frame 1: car:[0.327, 0.416, 0.623, 0.779], car:[0.004, 0.827, 0.99, 0.982], car:[0.961, 0.426, 1.0, 0.523], car:[0.76, 0.379, 0.966, 0.538]
Frame 2: car:[0.393, 0.427, 0.709, 0.777], car:[0.79, 0.387, 0.945, 0.553], car:[0.003, 0.825, 0.99, 0.98], car:[0.926, 0.434, 1.0, 0.549]
Frame 3: car:[0.518, 0.424, 0.849, 0.791], car:[0.834, 0.397, 0.994, 0.587], car:[0.007, 0.825, 0.983, 0.985]
Frame 4: car:[0.695, 0.542, 0.924, 0.777]
Frame 5:
Frame 6:
Frame 7: traffic light:[0.967, 0.525, 0.993, 0.613]

---

**Fixed QAs**
Question: what is the action of this vehicle?
Answer: The car turns right.
Question: Why does this vehicle behave in this way?
Answer: as the road is clear to turn.
Question: Predict the speed and turning angle of the ego vehicle.
Answer: Speed: 3.91, 3.1, 2.35, 2.92, 3.51, 4.24, 4.85, 5.22, 5.5;
          Turning angle: 0.0, -3.08, -5.98, -4.66, -2.91, 1.97, 7.02, 14.46, 20.04
**Conversations generated by ChatGPT**
User: What objects are present in the video, and how do they change throughout the frames?
AI: The video features various objects, including cars and a traffic light. As the video progresses, the positions and visibility of these objects change. In the initial frames, there are multiple cars. As the ego vehicle turns right, the surrounding cars gradually disappear from view. Towards the end of the video, a traffic light becomes visible.
User: How does the ego vehicle maneuver in the video?
AI: The ego vehicle starts driving straight and then makes a right turn. As the road becomes clear for turning, the ego vehicle accelerates and completes the turn safely.
User: What can we learn from the ego vehicle's interactions with the traffic and surrounding environment in this video?
AI: The ego vehicle's interactions with the traffic and surrounding environment demonstrate the importance of safe driving practices. The driver maintains a proper speed while navigating through traffic and pays attention to their surroundings, such as the positions of other vehicles and the traffic light. As they make the right turn, they ensure the road is clear before accelerating. These actions highlight the need for drivers to be aware of their environment, follow traffic rules, and prioritize safety when sharing the road with other users.

Table 1: Example of the instruction-tuning data sample. The upper part of this figure demonstrates input information, including video captions, control signals and object detection results obtained by YOLOv8. The lower part shows fixed QAs and conversations generated by ChatGPT. Refer to the appendix for detailed prompts.

ground truth object bounding boxes) and is prompted to generate conversations, descriptions, and reasoning. Currently, there is no visual instruction-following dataset tailored for autonomous driving purposes. Therefore, we create our own dataset based on BDD-X assisted by ChatGPT.

To address the aforementioned issue, ChatGPT is leveraged as a teacher to generate more conversations about the ego vehicle. The prompt generally follows the prompt design used in LLaVA. To enable ChatGPT to "see" the video, YOLOv8 (Reis et al., 2023) is implemented to detect commonly

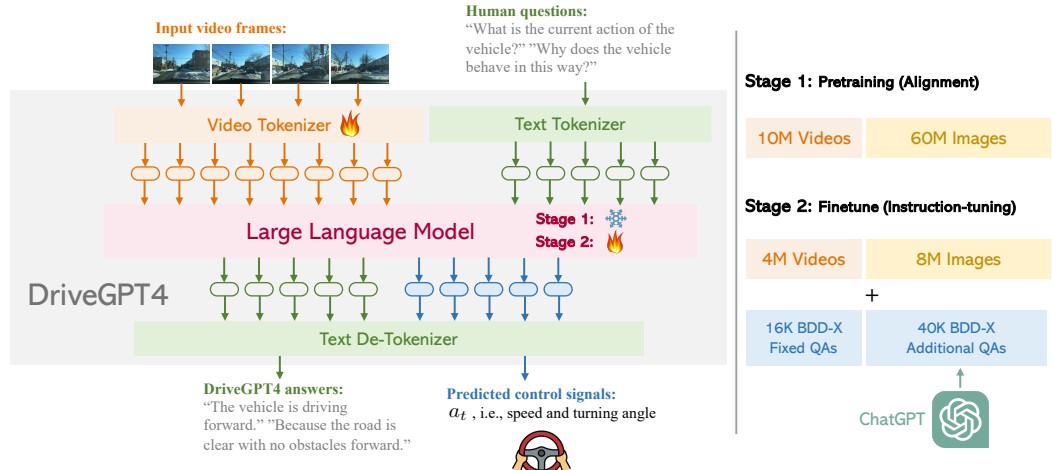

Figure 2: DriveGPT4 overview. DriveGPT4 is a comprehensive multimodal language model capable of processing inputs comprising videos, and texts. Video sequences undergo tokenization using a dedicated video tokenizer, while text and control signals share a common de-tokenizer. Following tokenization, the advanced language model can concurrently generate responses to human inquiries and predict control signals.

seen objects in each frame of the video (e.g., vehicles, pedestrians). Obtained bounding box coordinates are normalized and sent to ChatGPT as privileged information. In addition to object detection results, the video clip's ground truth control signal sequences and captions are also accessible to ChatGPT. Based on this privileged information, ChatGPT is prompted to generate multiple rounds and types of conversations about the ego vehicle, traffic lights, turning directions, lane changes, surrounding objects, spatial relations between objects, etc. Detailed prompt is provided in the appendix.

Finally, we collect 56K video-text instruction-following samples, including 16K fixed QAs and 40K conversations generated by ChatGPT. An example of a generated sample is shown in Tab. 1.

## 4 DRIVEGPT4

### 4.1 MODEL ARCHITECTURE

DriveGPT4 is a versatile multimodal LLM capable of handling various input types, including videos, and texts. Videos are uniformly sampled into a fixed number of images, and a video tokenizer based on Valley (Luo et al., 2023) is employed to convert video frames into text domain tokens. All generated tokens are concatenated and input into the LLM. In this paper, LLaMA 2 (Touvron et al., 2023b) is adopted as the LLM. Upon producing predicted tokens, a de-tokenizer decodes them to restore human languages. Drawing inspiration from RT-2 (Brohan et al., 2023), texts and control signals utilize the same text de-tokenizer, signifying that control signals can be interpreted as a language and effectively processed by LLMs. Decoded texts contain predicted signals in a fixed format. The overview architecture of DriveGPT4 is visualized in Fig. 2.

**Video tokenizer.** The video tokenizer is based on Valley (Luo et al., 2023). Let the input video frames be denoted as $V = [I_1, I_2, ..., I_N]$. For each video frame $I_i$, the pretrained CLIP visual encoder (Radford et al., 2021) is used to extract its feature $F_i \in \mathbb{R}^{257 \times d}$. The first channel of $F_i$ represents the global feature of $I_i$, while the other 256 channels correspond to patch features of $I_i$. For succinct representation, the global feature of $I_i$ is denoted as $F_i^G$, while the local patch features of $I_i$ are represented as $F_i^P$. The temporal visual feature of the entire video can then be expressed as:

$$T = F_0^G \oplus F_1^G \oplus ... \oplus F_N^G \tag{1}$$

where $\oplus$ denotes concatenation. The spatial visual feature of the whole video is given by:

$$S = \text{Pooling}(F_0^P, F_1^P, ..., F_N^P) \tag{2}$$

where $\text{Pooling}(\cdot)$ represents a pooling layer that convert $N$ features into a single feature tensor for memory efficiency. Ultimately, both the temporal feature $T$ and spatial feature $S$ of the video are

Human: This is a 8-frame video. The duration of this video is 2.1s. In this video, you are sitting in a vehicle on the road. What is the current action of this vehicle?
Assistant: The car slows down to a stop.
Human: What's driving the vehicle to behave in this way?
Assistant: since the light ahead became red.
Human: Forecast the speed and turning angle of the vehicle.
Assistant: Speed: 9.86 9.1 8.18 7.24 6.18 5.21 4.22 3.11 2.09
         Turning angle: 0.0 0.0 0.0 0.0 0.0 0.0 0.0 0.0 0.0

Table 2: Example of DriveGPT4 predictions. In this example, the input video is sampled into 8 frames. For concise visualization, only 4 frames are visualized.

projected into the text domain using a projector. The detailed structure of the tokenizer is depicted in Fig. 3.

**Text and control signals.** Inspired by RT-2 (Brohan et al., 2023), control signals are processed similarly to texts, as they belong to the same domain space. Control signals are directly embedded within texts during the process. The default LLaMA tokenizer is employed. In this study, the ego vehicle's speed $v = [v_1, v_2, ..., v_N]$ and turning angle $\Delta = [\Delta_1, \Delta_2, ..., \Delta_N]$ are considered as target control signals. The turning angle represents the relative angle between the current and initial frames. After obtaining predicted tokens, the LLaMA tokenizer is used to decode tokens back into texts. Predicted control signals are embedded in the output texts using a

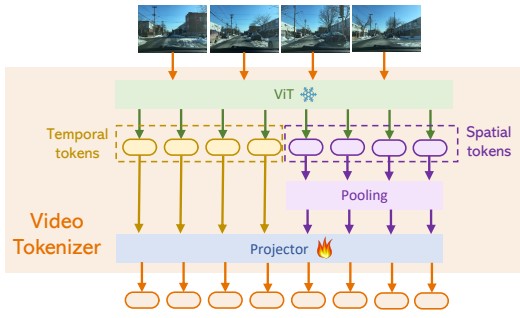

Figure 3: Architecture of the video tokenizer.

fixed format, allowing for easy extraction through simple post-processing. For control signals, DriveGPT4 should complete two sub-tasks: (1) estimate control signals of input video frames (i.e., $[v_1, v_2, ..., v_N]$ and $[\Delta_1, \Delta_2, ..., \Delta_N]$), and (2) predict control signals in the next step (i.e., $(v_{N+1}, \Delta_{N+1})$). An example illustrating the input and output of DriveGPT4 is presented in Tab. 2.

## 4.2 TRAINING

Consistent with previous LLM-related studies, DriveGPT4's training consists of two stages: (1) the pretraining stage, focusing on video-text alignment; and (2) the fine-tuning stage, aimed at training the LLM to answer questions related to end-to-end interpretable autonomous driving.

**Pretraining.** In line with LLaVA (Liu et al., 2023) and Valley (Luo et al., 2023), the model undergoes pretraining on 593K image-text pairs from the CC3M dataset and 100K video-text pairs from the WebVid-10M dataset (Bain et al., 2021). The pretraining images and videos encompass various topics and are not specifically designed for autonomous driving applications. During this phase, the CLIP encoder and LLM weights remain fixed. Only the video tokenizer is trained.

**Finetune.** In this stage, the LLM in DriveGPT4 is trained alongside the visual tokenizer for interpretable end-to-end autonomous driving. To enable DriveGPT4 to understand and process domain knowledge, it is trained with the 56K video-text instruction-following data generated in the previous section (Section 3). To enhance DriveGPT4's ability for visual understanding and question answering, 120K general instruction-following data generated by LLaVA and Valley are also utilized. Consequently, during the fine-tuning phase, DriveGPT4 is trained with 56K video-text instruction-following data for autonomous driving in conjunction with 120K general instruction-following data.

Table 3: Quantitative results of comparison experiments on the BDD-X dataset. We provide evaluation results on vehicle action description, action justification, and full sentences (i.e., combining description and justification). "B4", "C", "M" represent BLEU4, CIDEr and METETOR metric scores, respectively. "-" indicates that the results are not available.

| Method | Description | | | | Justification | | | | Full Sentence | | | |
|---|---|---|---|---|---|---|---|---|---|---|---|---|
| | B4↑ | C↑ | M↑ | ChatGPT↑ | B4↑ | C↑ | M↑ | ChatGPT↑ | B4↑ | C↑ | M↑ | ChatGPT↑ |
| ADAPT | 32.0 | 220.9 | 29.3 | 81.1 | 9.2 | 80.0 | 14.8 | 74.9 | 17.4 | 90.7 | 20.7 | 79.4 |
| Video-LLaMA | 1.7 | 3.9 | 7.5 | 42.2 | 4.7 | 2.8 | 9.3 | 33.9 | 3.9 | 8.9 | 3.1 | 40.3 |
| Valley | 2.2 | 10.3 | 9.2 | 66.3 | 5.0 | 2.8 | 9.4 | 67.2 | 5.7 | 2.9 | 10.4 | 63.3 |
| DriveGPT4 | 30.1 | 209.7 | 28.5 | **82.9** | 8.6 | 85.0 | 15.0 | **76.9** | 15.9 | 84.8 | 19.7 | **81.7** |

The former ensures that DriveGPT4 can be applied for interpretable end-to-end autonomous driving, while the latter enhances data diversity and the visual understanding ability of DriveGPT4.

## 5 EXPERIMENT

In this paper, DriveGPT4 focuses on interpretable end-to-end autonomous driving. With video frames and human questions as input, the method is required to predict interpretations in human language and control signals in the next step. Currently, except the BDD-X dataset, there are very few existing datasets that provide video clips captured by vehicle-mounted cameras with text interpretation and control signal annotations. Therefore, we mainly conduct evaluation experiments on the BDD-X dataset.

### 5.1 INTERPRETABLE AUTONOMOUS DRIVING

In this section, we evaluate DriveGPT4 and its baselines on interpretation generation, covering vehicle action description, action justification, and additional questions about vehicle status. ADAPT (Jin et al., 2023) serves as the state-of-the-art baseline work. Recent multimodal video understanding LLMs (Zhang et al., 2023; Luo et al., 2023) are also considered for comparison. All methods use 8-frame videos as input. Currently, DriveGPT4 does not take 32-frame videos as input like ADAPT considering the heavy memory consumption and inference speed, which is a limitation of this work.

**Evaluation Metrics.** To thoroughly assess the methods, we report multiple metric scores widely used in the NLP community, including BLEU4 (Papineni et al., 2002), METEOR (Banerjee & Lavie, 2005), and CIDEr (Vedantam et al., 2015). However, these metrics primarily measure word-level performance and do not account for semantic meaning, which may lead to unexpected evaluation results. Given ChatGPT's powerful reasoning ability, it is used to measure prediction quality and provide a

**Label:** The car is driving down the street.

**LLM pred:** The car is driving forward

(Bleu4: 0.49; METEOR: 0.30; **ChatGPT: 0.90**)

**ADAPT pred:** The car is driving down the street

(Bleu4: 1.0; METEOR: 1.0; **ChatGPT: 1.0**)

Figure 4: Example of evaluation metrics. Compared to conventional metrics, the score generated by ChatGPT is more reasonable and convincing for evaluation.

more reasonable score. ChatGPT is prompted to assign a numerical score between 0 and 1, with a higher score indicating better prediction accuracy. The detailed prompt for ChatGPT-based evaluation is available in the appendix. An example is shown in Fig. 4 for metric comparison. Compared to conventional metrics, the scores generated by ChatGPT provide a more reasonable and convincing basis for evaluation. Therefore, in this section, we primarily focus on the evaluation scores reported by ChatGPT. The reported ChatGPT scores are the mean result of three times of evaluations. Even though conventional metrics cannot effectively evaluate the results, we still report them for reference.

**Action Description and Justification.** Considering the cost and efficiency of the evaluation, DriveGPT4 is tested on 500 randomly sampled video clips from the BDD-X testing set. The goal is to predict vehicle action descriptions and justifications as closely as possible to the given labels. Evaluation results are displayed in Tab. 3. We report scores for 4 metrics pertaining to action description evaluation, action justification evaluation, and full sentence evaluation (i.e., combining

Figure 5: Conversations generated by DriveGPT4. In these examples, it is tested on the BDD-X testing set.

description and justification). Based on more convincing scores reported by ChatGPT, the results indicate that DriveGPT4 achieves superior performance compared to the previous state-of-the-art (SOTA) method, ADAPT.

**Additional Question Answering.** The above vehicle action description and justification have relatively fixed formats. To further evaluate the interpretable ability and flexibility of DriveGPT, additional questions are generated following section 3. A hundred randomly sampled video clips in the BDD-X testing set are used for question generation. Compared with action descriptions and justifications, these questions are more diverse and flexible. The evaluation results are shown in Tab. 4. ADAPT cannot answer additional questions except for the vehicle action description and jus-

Table 4: Quantitative results of comparison experiments on additional question answering. The model is required to answer questions generated by ChatGPT. "B4", "C", "M" stand for BLEU4, CIDEr and METE-TOR, respectively.

| Method | B4↑ | C↑ | M↑ | ChatGPT↑ |
|---|---|---|---|---|
| ADAPT | - | - | - | - |
| Video-LLaMA | 2.9 | 5.7 | 10.2 | 27.7 |
| Valley | 5.0 | 11.3 | 11.0 | 43.2 |
| DriveGPT4 | **21.7** | **52.4** | **22.0** | **79.9** |

tification. Previous video understand LLMs can answer these questions but they do not learn autonomous driving domain knowledge. Compared with all baselines, DriveGPT4 presents superior results, demonstrating its flexibility.

## 5.2 END-TO-END CONTROL

In this section, we evaluate DriveGPT4 and its baselines for open-loop control signal prediction, specifically focusing on speed and turning angle. All methods are required to predict control signals for the next single frame based on sequential input.

Table 5: Quantitative results of comparison experiments on the BDD-X dataset. We provide evaluation results on vehicle action description, justification, and full sentences (i.e., combining description and justification).

| Method | Speed (m/s) | | | | | Turning angle (degree) | | | | |
|---|---|---|---|---|---|---|---|---|---|---|
| | RMSE↓ | $A_{0.1}$ ↑ | $A_{0.5}$ ↑ | $A_{1.0}$ ↑ | $A_{5.0}$ ↑ | RMSE↓ | $A_{0.1}$ ↑ | $A_{0.5}$ ↑ | $A_{1.0}$ ↑ | $A_{5.0}$ ↑ |
| ADAPT | 3.11 | 9.56 | 24.77 | 37.07 | **90.39** | 11.99 | 27.93 | **66.83** | **75.13** | **89.45** |
| DriveGPT4 | **2.92** | **16.73** | **30.66** | **42.97** | 88.99 | **10.72** | **40.01** | 53.26 | 62.14 | 83.03 |

**Evaluation Metrics.** Following previous works on control signal prediction, we use root mean squared error (RMSE) and threshold accuracies ($A_\tau$) for evaluation. $A_\tau$ measures the proportion of test samples with prediction errors lower than $\tau$. For a comprehensive comparison, we set $\tau$ with multiple values: $\{0.1, 0.5, 1.0, 5.0\}$.

**Quantitative Results.** After removing samples with incorrect control signal labels, all other samples in the BDD-X testing set are used for control evaluation. The quantitative results for the previous state-of-the-art (SOTA) method ADAPT and DriveGPT4 are shown in Tab. 5. DriveGPT4 achieves superior control prediction results.

## 5.3 QUALITATIVE RESULTS.

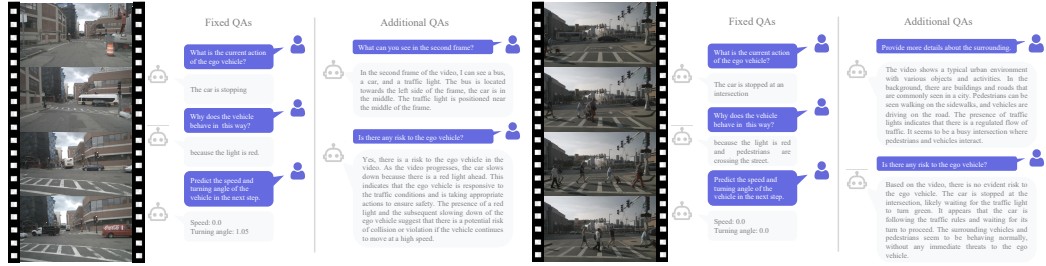

Figure 6: Conversations generated by DriveGPT4. In these examples, DriveGPT4 is generalized to the NuScenes dataset (Caesar et al., 2020) in a zero-shot manner.

Multiple qualitative results are provided for intuitive comparison. First, two examples from the BDD-X testing set are visualized in Fig. 5. Then, to verify the generalization ability of DriveGPT4, we apply DriveGPT4 to the NuScenes dataset (Caesar et al., 2020) for zero-shot conversation generation in Fig. 6. Finally, we try DriveGPT4 on video games to further test its generalization ability. An example is illustrated in Fig. 7.

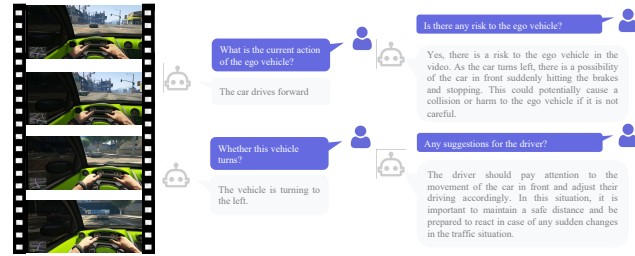

Figure 7: Conversations generated by DriveGPT4. In this example, DriveGPT4 is generalized to video games in a zero-shot manner.

## 5.4 ABLATION STUDIES

Table 6: Quantitative results of comparison experiments on the BDD-X dataset. We provide evaluation results on vehicle action description, justification, and full sentences (i.e., combining description and justification).

| Method | BDD-X Questions | | | | ChatGPT Questions | | | | Speed | | |
|---|---|---|---|---|---|---|---|---|---|---|---|
| | B4↑ | C↑ | M | ChatGPT↑ | B4↑ | C↑ | M↑ | ChatGPT↑ | RMSE↓ | $A_{0.1}$ ↑ | $A_{1.0}$ ↑ |
| No fixed-QA | 1.0 | 2.4 | 1.3 | 67.7 | 21.6 | 49.5 | 21.7 | 81.3 | - | - | - |
| No ChatGPT QA | 14.9 | 76.3 | 17.9 | 81.5 | 0.0 | 2.9 | 2.3 | 29.6 | 2.97 | 17.90 | 41.86 |
| No global features | 9.7 | 49.7 | 9.6 | 50.7 | 12.0 | 22.3 | 7.9 | 38.4 | 8.96 | 6.70 | 19.04 |
| DriveGPT4 | 15.9 | 84.8 | 19.7 | 81.7 | 21.7 | 52.4 | 22.0 | 79.9 | 2.92 | 16.73 | 42.97 |

In this paper, several ablation studies are conducted to validate the proposed designs, and the results are provided in Tab. 6. First, by removing either fixed question-answer pairs or ChatGPT-generated conversations during fine-tuning, a decrease in corresponding performance is observed, highlighting the significance of including all visual-text instruction-following data. QA pairs generated by Chat-GPT enable DriveGPT4 to answer human questions in more flexible patterns, and slightly enhance the QA ability of BDD-X questions. Additionally, when DriveGPT4 is tested without the global feature in the video tokenizer, it fails to capture the temporal information of the input video clip, so it can only make predictions based on static visual information, and degraded performance is observed. Thus, any changes to the design of DriveGPT4 would negatively impact its versatile QA capabilities for interpretable end-to-end autonomous driving.

## 6 CONCLUSION

This paper presents DriveGPT4, an interpretable end-to-end autonomous driving system using multimodal LLM. A new dataset for autonomous driving interpretation is developed with the assistance of ChatGPT and employed to fine-tune DriveGPT4, enabling it to respond to human inquiries about the vehicle. DriveGPT4 utilizes input videos and texts to generate textual responses to questions and predict control signals for vehicle operation. It outperforms baseline models in various tasks such as vehicle action description, action justification, general question answering, and control signal prediction. Moreover, DriveGPT4 exhibits generalization ability through zero-shot adaptation.

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

## A  DATA PROCESSING

In the data processing part, we generate three question sets for fixed question answering, i.e., $Q_a$ for action description, $Q_j$ for action justification and $Q_c$ for control signals. The detailed question sets are shown in Tab. 7, Tab. 8 and Tab. 9, respectively.

The detailed prompt to generate conversations by ChatGPT is shown in Tab. 10.

> What is the current action of this vehicle?
> What is the vehicle doing right now in this video?
> What action is the vehicle performing in this video at the moment?
> Can you describe the vehicle's current activity in this video?
> What's happening with the vehicle in this video right now?
> At this moment in the video, what is the vehicle engaged in?
> What can you observe the vehicle doing in this video currently?
> How is the vehicle behaving at this point in the video?
> What is the ongoing action of the vehicle in the video?
> In this video, what action is the vehicle involved in at present?
> Describe the current state of the vehicle in this video.

Table 7: Question set $Q_a$.

> Why does this vehicle behave in this way?
> What is the reason behind this vehicle's behavior?
> Can you explain the cause of this vehicle's actions?
> What factors contribute to the way this vehicle is behaving?
> What's the rationale behind this vehicle's behavior?
> Why is the vehicle acting in this particular manner?
> What prompted the vehicle to behave like this?
> What circumstances led to this vehicle's behavior?
> What is the underlying cause of this vehicle's actions?
> For what reason is the vehicle exhibiting this behavior?
> What's driving the vehicle to behave in this way?

Table 8: Question set $Q_j$.

> Predict the speed and turning angle of the vehicle in the next frame.
> Foresee the speed and turning angle of the vehicle in the following frame.
> Anticipate the speed and turning angle of the vehicle in the subsequent frame.
> Estimate the speed and turning angle of the vehicle in the next frame.
> Project the speed and turning angle of the vehicle in the upcoming frame.
> Forecast the speed and turning angle of the vehicle in the ensuing frame.
> Envision the speed and turning angle of the vehicle in the next frame.
> Expect the speed and turning angle of the vehicle in the following frame.
> Presume the speed and turning angle of the vehicle in the subsequent frame.
> Prognosticate the speed and turning angle of the vehicle in the next frame.
> Calculate the speed and turning angle of the vehicle in the upcoming frame.

Table 9: Question set $Q_c$.

## B  EVALUATION SCORES GENERATED BY CHATGPT

The prompt for evaluation score generation is demonstrated in Tab. 11. For each question-answer pair, we embed them in the prompt texts and send them to ChatGPT. ChatGPT first outputs a numerical number ranging from 0 to 1, and then provides explanations to the score. An example showing scores and explanations is provided in Fig. 8.

There is a 8-frame video recording a drive driving a vehicle. {BDD-X captions}. There are some exclusive privilege information, but you cannot mention them in your generated question answering. 1. Objects in each frame of the video: {objects}; 2. The speed (m/s) of the vehicle in each frame :{speed}. The turning angle (degree) of the vehicle in each frame :{turning angle}.

Design a conversation between you and a person asking about this video. The answers should be in a tone that a visual AI assistant is seeing the video and answering the question. Ask diverse questions and give corresponding answers.

Include questions asking about the visual content of the video, including the ego vehicle, traffic light, turning direction, lane change, surrounding objects, objects spatial relations, etc. Only include questions that have definite answers:
(1) one can see the content in the video that the question asks about and can answer confidently; (2) one can determine confidently from the video that it is not in the video. Do not ask any question that cannot be answered confidently.
Do not contain specific numbers in the questions, e.g., normalized coordinates, speed value, turning angle.

Also include complex questions that are relevant to the content in the video, for example, asking about background knowledge of the objects in the video, asking to discuss about events happening in the video, etc. Again, do not ask about uncertain details. Provide detailed answers when answering complex questions. For example, give detailed examples or reasoning steps to make the content more convincing and well-organized. You can include multiple paragraphs if necessary.

The conversation should be 3 turns. Make the answer concise and accurate.

Table 10: Prompts for visual instruction generation. ChatGPT can access privileged information like ground truth BDD-X captions, object boxes, vehicle speeds and turning angles.

Now there are some descritions about a driver driving a vehicle. The ground truth description is: {GT label}. The description generated by deep learning model is: {Prediction}.

Give me an evaluation score about the predicted description. The score should range from 0 to 1. Larger score means better description. The score should be a float number with 2 decimal places. For example, 0.51, 0.99, 0.00, 0.76, etc.

You should first give me the score number, and then provide explanations for your score number.

Table 11: Prompts for response text evaluation. Compared with conventional metrics, the score generated by ChatGPT is more reasonable and convincing.

**Label:** The car is driving down the street.

**LLM pred:** The car is driving forward

Bleu4: 0.49
METEOR: 0.30
**ChatGPT:** 0.90

**Score reasoning:** The generated description "The car is driving forward" captures the essential action of the car moving in a direction but lacks the contextual detail of the car driving "down the street" as the ground truth description provides. However, the prediction is still reasonably accurate, which is why it receives a score of 0.90.

**ADAPT pred:** The car is driving down the street

Bleu4: 1.0
METEOR: 1.0
**ChatGPT:** 1.0

**Score reasoning:** The predicted description is identical to the ground truth description: "The car is driving down the street." Since the prediction matches the ground truth perfectly, the evaluation score is 1.00.

Figure 8: Evaluation results comparison. In this example, we provide texts generated by ChatGPT to explain its evaluation scores.

## C   PREDICTION CONSISTENCY

Here are texts describing a vehicle action and its control signals.

Positive examples:
1. [The car is stopped",”because the light is red.",”Speed: 0.21; Turning angle: 0.0"].
2. [”The car is turning left",”because the light is green.",”Speed: 5.47; Turning angle: -89.59"].

Negative examples:
1. [The car is turning left",”because the light is green.",”Speed: 5.47; Turning angle: 89.59"]: The car is tunring left, so that the turning angle should be negative, thus the control and the text are not consistent.
2. [The car is stopped",”because the light is red.",”Speed: 10.21; Turning angle: 0.0"]: The car should be stopped, but the speed is very high, which is not correct.

Now, please provide a score ranging from 0-1 to measure the consistency of the control and text of the following case. Note that you should only provide a score figure, and no other text is allowed.
{Predictions to be evaluated}

Table 12: Prompts for prediction consistency evaluation.

**Predictions:** ["The car is driving forward",
"because the light is green and there is no traffic impeding it.",
"Speed: 13.16; Turning angle: 89.05"]

**ChatGPT consistent score:** 0.20

**Score reasoning:** In this case, I would give a score of 0.2 to the consistency of the control and text. The car is driving forward because the light is green and there is no traffic impeding it, which is consistent with the given information. However, a turning angle of 89.05 indicates a sharp turn, which contradicts the statement of the car driving forward.

**Predictions:** ["The car is turning left",
"because the light is green.",
"Speed: 5.47; Turning angle: -2"]

**ChatGPT consistent score:** 0.90

**Score reasoning:** In this case, I would give a score of 0.9 to the consistency of the control and text. The car is turning left because the light is green, which is consistent with the given information. The speed of 5.47 and a turning angle of -2 indicate a controlled and gradual left turn.

Figure 9: Examples of consistency evaluation. In these examples, we prompt ChatGPT to explain its evaluation scores.

To measure whether DriveGPT4 makes consistent control signal and text response predictions, ChatGPT is utilized for consistency evaluation. ChatGPT is prompted to generate a score ranging from 0 to 1 to measure the consistency between generated text answers and predicted control signals. The detailed prompt for ChatGPT is demonstrated in Tab. 12. Consistency evaluation examples are shown in Fig. 9. The evaluation results are listed in Tab. 13. The consistency scores of ADAPT and DriveGPT4 are similar. In most situations, both ADAPT and DriveGPT4 can have consistent predictions.

Table 13: Evalution results of prediction consistency.

| Method | ChatGPT↑ |
| --- | --- |
| ADAPT | 79.40 |
| DriveGPT4 | 80.69 |

## D  ADDITIONAL QUALITATIVE RESULTS

We provide more qualitative results of DriveGPT4 for interpretable end-to-end autonomous driving in Fig. 10 to 13.

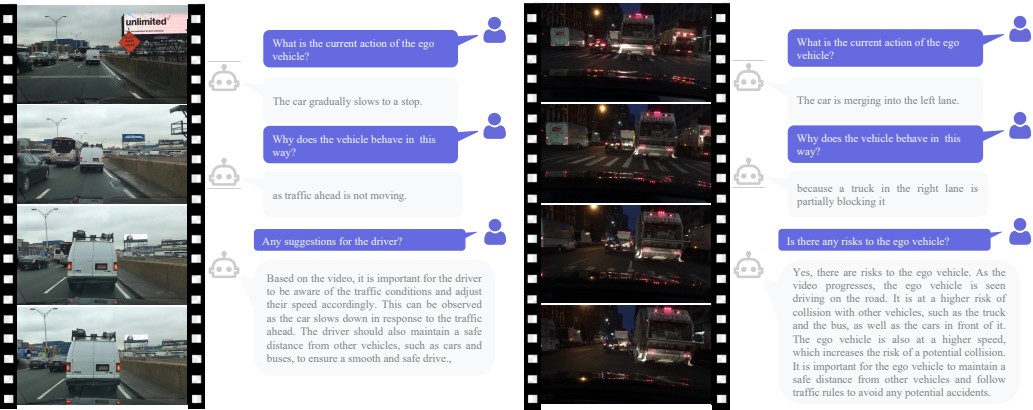

Figure 10: Conversations generated by DriveGPT4. In these examples, it is tested on the BDD-X testing set.

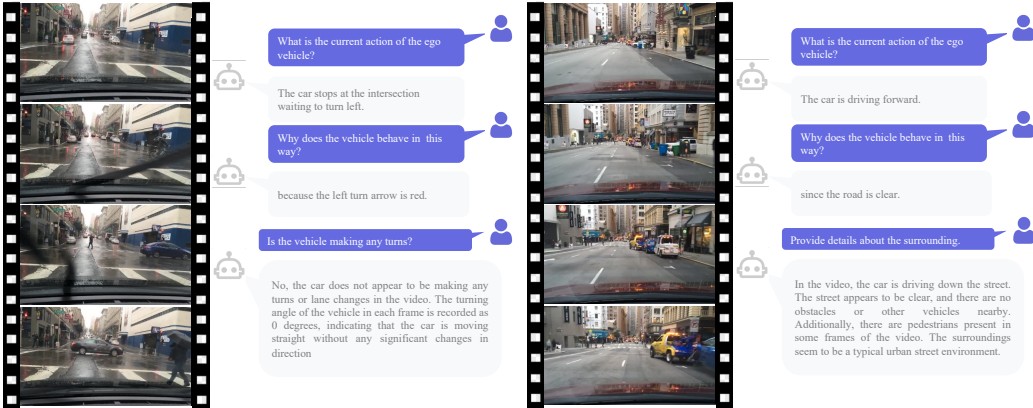

Figure 11: Conversations generated by DriveGPT4. In these examples, it is tested on the BDD-X testing set.

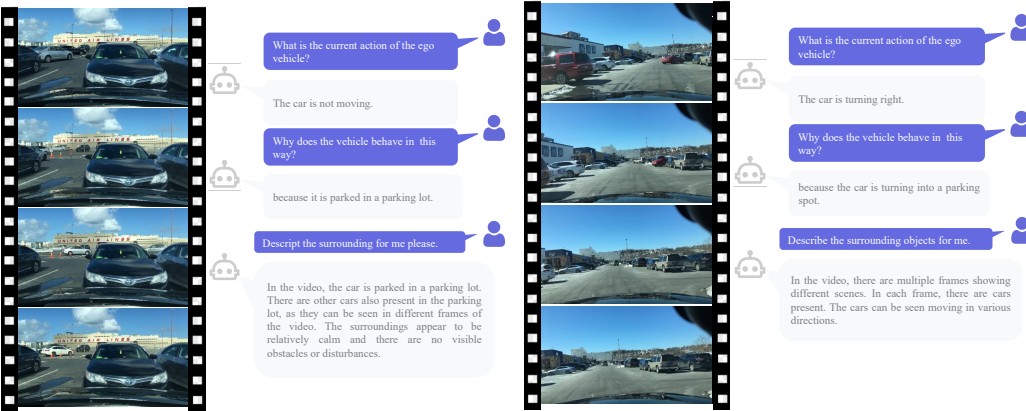

Figure 12: Conversations generated by DriveGPT4. In these examples, it is tested on the BDD-X testing set.

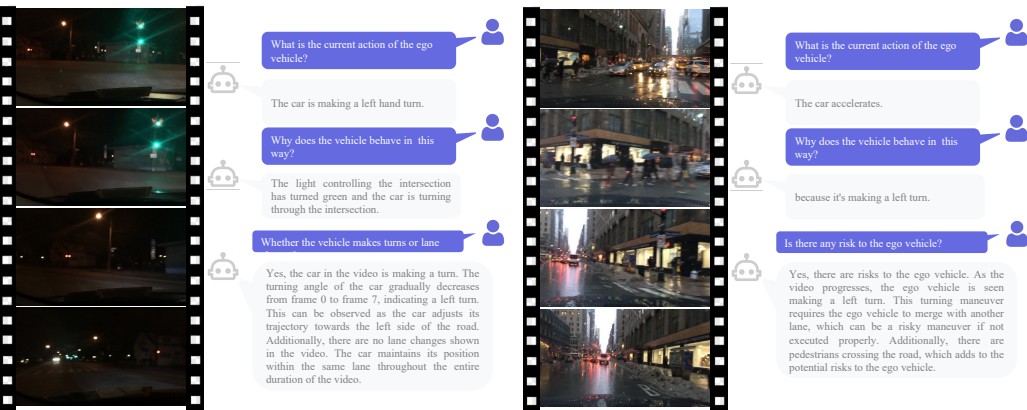

Figure 13: Conversations generated by DriveGPT4. In these examples, it is tested on the BDD-X testing set.

