# OpenReview forum: "DriveGPT4: Interpretable End-to-end Autonomous Driving via Large Language Model"
_ICLR.cc/2024/Conference — Submitted to ICLR 2024_

### Official Review · Reviewer_2Geq · 2023-10-28

**Soundness:** 2 fair
**Presentation:** 3 good
**Contribution:** 2 fair
**Rating:** 5
**Confidence:** 4

**Summary:**

Despite the rapid progress of autonomous driving, these systems do not interact with human in natural language and the dominant approach decomposes the problem into perception, prediction and control. Depart from these approaches, this paper presents DriveGPT4, an interpretable end-to-end autonomous driving system leveraging large language models. DriveGPT4 takes as input a video sequence captured by a front-view RGB camera, along with the vehicle’s historical control signals. It predicts the control signal for the next step and can provide natural language responses, such as describing the vehicle’s actions and explaining the reasoning behind its behavior.

To train DriveGPT4 to communicate with natural language, the paper follows LLaVA (Liu et al., 2023) and creates a visual instruction tuning dataset based on the BDD-X dataset (Kim et al., 2018) using ChatGPT. The model DriveGPT4 is based on Valley(Luoetal.,2023) and fine-tuned on the created dataset. The model is mainly compared with ADAPT (Jin et al., 2023), Action-aware driving caption transformer.

**Strengths:**

1. Leveraging large language model LLaMA 2, the paper develops a multi-modal action model which takes video, text, historical control signals as input and outputs the control signals for the next step and can respond in natural language to explain driving actions and behavior.

2. It creates a visual instruction tuning dataset based on the BDD-X dataset (Kim et al., 2018) using ChatGPT.

3. The model is compared with ADAPT (Jin et al., 2023) on question answering and control signal prediction tasks.

**Weaknesses:**

1. The model relies on behavior cloning for end-to-end driving. This is the first end-to-end method tried out. The limitation is very well-known, e.g. can not handle distribution drift. For more information, please see

End-to-end Autonomous Driving: Challenges and Frontiers
Li Chen, Penghao Wu, Kashyap Chitta, Bernhard Jaeger, Andreas Geiger, Hongyang Li

2. The paper relies on ChatGPT to evaluate on vehicle action description, action justification, question and answering. ChatGPT can exhibit  well-known bias such as position bias, style bias. Given the data is generated by ChatGPT and the baseline ADAPT does not use ChatGPT, ChatGPT evaluation could be very well biased in favor of DriveGPT4 simply because it prefers its own style.

3. The action prediction task should be evaluated with strong end-to-end autonomous driving baselines.

**Questions:**

I have a number of additional questions.

1. For architectural choices, why not just fine-tune Video-LLaMA with the visual instruction tuning dataset based on the BDD-X dataset? Do you employ position encoding for video tokens?

2.  In ablation study, it will be great to see the contribution of global features by removing F0^G ⊕ F1^G ⊕ ... ⊕ FN^G.

3. The paper uses Yolo-8 for object detection, I wonder if the authors can comment on whether open-set detectors like grounding DINO could be better.

4. The training stages, methods, datasets, and tasks are scattered. It would help if they can be summarized in a table. Furthermore, the architecture figure should have information on the training stages, which part is frozen or trainable.

5. DriveGPT4 dataset is based on BDD-X dataset. DriveLLM2023 (https://github.com/OpenDriveLab/DriveLM) dataset augments NuScene with QA + Scene Description from the perspective of perception, prediction and planning with Logic. Can the model be evaluated on this dataset as well? Specifically how well DriveGPT4 performs on high level planning decisions compared with DriveLLM?

**Details Of Ethics Concerns:**

Since this is a research prototype, there is not much ethics concerns.

---

> ### Author Response · Authors · 2023-11-21
> **Response to Reviewer 2Geq**
>
> We thank the reviewer for detailed comments and suggestions. Our responses to the comments and questions of the reviewer are listed below.
>
>
>
> **1.The model relies on behavior cloning for end-to-end driving. This is the first end-to-end method tried out. The limitation is very well-known, e.g. can not handle distribution drift. For more information,**
>
> Thanks for the information. We have citated this paper in our revised manuscript. At this stage, DriveGPT4 is for open-loop vehicle control. The behavior cloning issue mentioned in this question should be carefully studied in the close-loop control, which is the research direction in the next stage.
>
> **2. Given the data is generated by ChatGPT and the baseline ADAPT does not use ChatGPT, ChatGPT evaluation could be very well biased in favor of DriveGPT4 simply because it prefers its own style.**
>
> The fixed QAs are not generated by ChatGPT, and they are directly from the BDD-X dataset. For fixed QAs, ChatGPT is used to generate more patterns of the question, but the answer stays fixed and not altered. During evaluation, the generated answers are not in ChatGPT style, but in BDD-X style. Thus ChatGPT evaluation does not show biased favor for this, and the results can be reliable.
>
> In addition, compared with conventional evaluation metrics, the ChatGPT metric is much more powerful since it can measure complicated semantic meanings/logic/etc. Evaluation with ChatGPT/GPT4 is commonly conducted in many recent papers about multimodal LLM [1,2]. Therefore, leveraging ChatGPT for evaluation should be a good way for comparison.
>
>
>
> **3. The action prediction task should be evaluated with strong end-to-end autonomous driving baselines.**
>
> DriveGPT4 is proposed for interpretable end-to-end autonomous driving, thus it should be able to predict the text explanations as well as vehicle actions at the same time. ADAPT is the previous SOTA for this problem, thus DriveGPT4 is mainly compared with ADAPT, instead of other baselines that only focus on either interpretability or end-to-end vehicle control.
>
> **4. why not just fine-tune Video-LLaMA with the visual instruction tuning dataset based on the BDD-X dataset? Do you employ position encoding for video tokens?**
>
> Yes, positional encoding is used in DriveGPT4 for video frames. DriveGPT4 is basaed on a video understanding framework Valley, which is similar to Video-LLaMA but presents better video understanding ability. Thus, DriveGPT4 follows Valley for model design and tuning.
>
> **5. In ablation study, it will be great to see the contribution of global features by removing**
>
> Thanks for the suggestion. The results have been added into the ablation study.  Please refer to Table. 6 in the revised manuscript.
>
> **6. The paper uses Yolo-8 for object detection, I wonder if the authors can comment on whether open-set detectors like grounding DINO could be better.**
>
> We tried the mentioned open-set detectors for object detection during data generation. But since we only care about limited object classes, e.g., vehicle, pedestrians, traffic light, we did not obvious notable difference between YOLO and open-set detectors. YOLO is already effective and efficient enough for our specific applications.
>
> **7. The training stages, methods, datasets, and tasks are scattered. It would help if they can be summarized in a table. Furthermore, the architecture figure should have information on the training stages, which part is frozen or trainable.**
>
> Thanks for the suggestion. We have modified the manuscript accordingly. Please refer to the revised manuscript for details.
>
> **8. Can the model be evaluated on this dataset as well? Specifically how well DriveGPT4 performs on high level planning decisions compared with DriveLLM?**
>
> DriveLLM does not release any code or code at this stage yet. As long as DriveLM releases their data, we will adapt our model on it and conduct additional evaluations.
>
> **References**
>
> [1] Liu, Haotian, et al. "Visual instruction tuning." NIPS, 2023.
>
> [2] Luo, Ruipu, et al. "Valley: Video Assistant with Large Language model Enhanced abilitY." arXiv:2306.07207, 2023.

---

### Official Review · Reviewer_EsLd · 2023-10-29

**Soundness:** 1 poor
**Presentation:** 3 good
**Contribution:** 2 fair
**Rating:** 3
**Confidence:** 5

**Summary:**

This paper proposes a multimodal LLM-based interpretable end-to-end autonomous driving system. Based on the BDD-X dataset, this paper constructs an instruction-tuning dataset comprised of fixed-form QAs and free-form QAs & conversations with ChatGPT. The model is comprised of a video tokenizer to extract video input features and an LLM to process the multi-modal inputs and make textual responses. The model also predicts control signals in the text format. The training has an alignment stage and fine-tuning stage similar to other multi-modal LLMs. The experiments evaluate the interpretability on action description&justification and QAs tasks and the end-to-end control ability on speed and turning angle prediction, which shows the superiority of the proposed model.

**Strengths:**

1. This paper constructs an instruction dataset for AD and shares the pipeline for its construction, which benefits the community for future research.
2. This work shows promising results in letting multi-modal LLMs understand autonomous driving scenarios.
3. The presentation is clear and easy to follow.

**Weaknesses:**

1. As the model is an end-to-end autonomous driving model, the ability to make driving plans or control predictions is critical. However, from the model design and the ablation experiments, it seems that the model might just simply extrapolate the input control signals. The authors should also include more possible baselines for this task (for example, a simple transformer with the same input as the proposed model which is directly trained on the prediction task). Besides, only predicting the speed and steer angles might not be enough to claim it as an end-to-end autonomous driving model.
2. Although the paper claims that it is interpretable end-to-end autonomous driving, there are no explicit constraints between the model's textual explanation and its predicted action during training. And there are no experiments to validate this as well.
3. The paper claims that the proposed model still has the general multi-modal conversation ability. Then the author should evaluate it on the general multi-modal benchmarks to validate this.
4. For the experiments about interpretability, the proposed method shows consistently inferior performance in terms of standard metrics. It is true that the classic metrics might have certain problems, but so does the ChatGPT-based evaluation. As the training data is partially generated by ChatGPT, ChatGPT might prefer the responses similar to its own generated ones during evaluation.

**Questions:**

Why do you think removing the historical control signals as inputs makes the performance much worse if the model really learns the driving policy?

---

> ### Author Response · Authors · 2023-11-21
> **Response to Reviewer EsLd**
>
> We thank the reviewer for detailed comments and suggestions. Our responses to the comments and questions of the reviewer are listed below.
>
>
>
> **1. However, from the model design and the ablation experiments, it seems that the model might just simply extrapolate the input control signals. The authors should also include more possible baselines for this task**
>
> Thanks for this good question. In the revised manuscript, history information is removed from the input. Now DriveGPT4 makes prediction only based on video frames and input human questions. Please refer to Table.3 in the revised manuscript for the new results. From the results, we can see that DriveGPT4 still outperforms the previous SOTA ADAPT (8-frames for fair comparison).
>
> **2. Although the paper claims that it is interpretable end-to-end autonomous driving, there are no explicit constraints between the model's textual explanation and its predicted action during training**
>
> Currently, there are no constraints on generated texts and actions. Because based on our experiment results, the predicted texts well align with predicted actions. Because both texts and actions are predicted based on the input video frames and questions, if the LLM has incorrect understanding about the input video clip, it would generate incorrect answers and incorrect actions at the same time. At this stage, we did not observe obvious inconsistency between the predicted texts and actions, so we did not cast addition constriants (losses) on them.
>
> **3. The paper claims that the proposed model still has the general multi-modal conversation ability. Then the author should evaluate it on the general multi-modal benchmarks to validate this.**
>
> Sorry for the overclaim. In the original paper, we meant that DriveGPT4 is more flexible than ADAPT to answer questions in more patterns raised by humans. Thus, this ability is the additional question-answering ability, which is evaluated in Table 4 of the manuscript. DriveGPT4 is specially designed for autonomous driving scenarios, and it is not trained for other general multimodal problems, e.g., image understanding, video understanding, etc. Overclaims have been removed from the revised manuscript.
>
> **4. As the training data is partially generated by ChatGPT, ChatGPT might prefer the responses similar to its own generated ones during evaluation.**
>
> The answers to fixed QA pairs are directly from BDD-X dataset, and the model is trained to answer in BDD-X label style instead of ChatGPT style. Besides, no matter what style of the generated response, ChatGPT compares it with the BDD-X label, so evaluation bias should not happen here.
>
> Compared with conventional evaluation metrics, the ChatGPT metric is much more powerful since it can measure complicated semantic meanings/logic/etc. Evaluation with ChatGPT/GPT4 is commonly conducted in many recent papers about multimodal LLM [1,2]. Therefore, leveraging ChatGPT for evaluation should be a good way for comparison.
>
> **References**
>
> [1] Liu, Haotian, et al. "Visual instruction tuning." NIPS, 2023.
>
> [2] Luo, Ruipu, et al. "Valley: Video Assistant with Large Language model Enhanced abilitY." arXiv:2306.07207, 2023.

---

> > ### Comment · Reviewer_EsLd · 2023-11-22
> > **Response to Authors**
> >
> > I appreciate the authors' effort in providing additional experiments and explanations during the rebuttal period. I still have few concerns.
> > 1. For the new experiments without history information, you modified the baseline ADAPT-32 to ADAPT-8. However, as the default setting of ADAPT is 32 frames, you should also include the comparison where your model takes in 32 frames as input (if it is impossible due to the computational cost, you should mention this as it is the limitation of this architecture).
> > 2. Certain evaluation metrics should be designed to validate this consistency.
> > 3. As you also include instruction-following data generated by LLaVA and Valley for training, it would be helpful to provide the evaluation results on general multi-modal benchmarks to see whether adding additional driving domain data would impede the general multi-modal ability.

---

> > > ### Author Response · Authors · 2023-11-23
> > > **Response to Reviewer EsLd**
> > >
> > > Thanks for your prompt feedback and comments. For your concerns:
> > >
> > > 1. Currently DrvieGPT4 cannot handle that heavy input with 32 frames as input. Thus we claim this as a limitation of this project at the current step. We have added corresponding clarifications about this in section 5.1 of the revised manuscript.
> > >
> > > 2. We leverage ChatGPT to measure the consistency of the predicted control signals and text answers. The results are provided below (the evaluation is run two times and takes the mean).
> > > | Method      | ChatGPT Score |
> > > | ----------- | ----------- |
> > > | ADAPT     | 79.40       |
> > > | DriveGPT4   | 80.69        |
> > >
> > >  The consistency scores of ADAPT and DriveGPT4 are similar. Under most circumstances, both ADAPT and DriveGPT4 can have consistent predictions, either generating both correct control signals and texts or incorrect control signals and texts. Due to the page limit, The detailed prompt and evaluation results about prediction consistency are provided in the appendix section C. Please refer to the newly uploaded revised manuscript.
> > >
> > > 3. Thanks for the suggestion. We agree with the reviewer that adding evaluation for multimodal QA can make the paper more solid. Since the rebuttal time is limited, we will conduct a corresponding comparison in the near future. Here we apologize for giving the response and revision at the end of the rebuttal period, since it takes a lot of time and resources to refine the model structure and conduct corresponding experiments.

---

### Official Review · Reviewer_yQKg · 2023-10-30

**Soundness:** 2 fair
**Presentation:** 3 good
**Contribution:** 2 fair
**Rating:** 5
**Confidence:** 4

**Summary:**

This paper proposes to apply large language models in the end-to-end autonomous driving domain. First of all, it constructs a question-answering dataset based on the BDD-X dataset. By providing ChatGPT with fixed question-answer pairs, and more privileged information from BDD-X, it generates more conversations, and descriptions about the scene or the reasoning process. The proposed DriveGPT4 model takes video as input, tokenized videos, questions and past control signals, and then finetunes LLaMA 2 to decode the required answers and control signals. The training process also involves image-text pairs from other domains to facilitate out-of-domain questions. The experiments are mainly compared with ADAPT on QA and action prediction metrics.

**Strengths:**

- The paper is basically clearly written and easy to understand.
- The authors demonstrate that the LLM-based method has the zero-shot generalization ability to other datasets. It is a good point to involve LLM or large-scale training for images and videos.

**Weaknesses:**

Multiple overclaims and issues about the soundness.

- *Important:* There are multiple sentences saying that "constrained by the limited capacity of smaller language models, they can only address predefined human questions and provide inflexible answers, hindering their widespread application in real-world scenarios". The reviewer admits that LLMs can answer out-of-domain questions, but is still wondering if the limitation lies in the LLM. I will explain this point based on the experiments.
  - The improvement of the fixed question-answering experiment is limited (82.1->82.4, especially considering this is evaluated by ChatGPT with great uncertainty). Meanwhile, LLMs take much longer inference time to output the much longer answers compared to small models and the original QAs. The computational complexity is not that important at the current stage but is still valuable for illustration.
  - The additional question-answering experiment demonstrates the effectiveness of DriveGPT4 which is finetuned on the generated data. Then how about fintuning ADAPT and other works on the data? The direct comparison seems unfair.
  - For the control prediction task, DriveGPT4 takes historical control signals as input while others do not. It can be observed from the ablation study, that the results of 'No control signals' are close to ADAPT-32. The contribution comes from the history information, rather than LLM, which is unfair.
  - It is an open question if flexible answers can help real-world applications of autonomous driving.
- It is not the **first** work on interpretable end-to-end autonomous driving. The authors have already listed several in the related works. Even language-based interpretability is not accurate as there are multiple types of interpretability. In my opinion, it is fair to say LLM-based.
- From the very beginning, is **interpretability** indeed the unsolved problem that hinders the commercialization and development of autonomous driving? As described in the first paragraph of the introduction, the popular modular-based methods' issue is not the interpretability.

Technical contribution is limited. There are contributions of instruction tuning for the dataset generation. The video tokenization part is based on Valley and the action part is very close to existing works such as RT-2.

**Questions:**

- Why not use ground truth from the BDD dataset? The detection results from YOLOv8 could be inaccurate. I am wondering if there is a truck in the provided figure (Table 1).
- How bounding box coordinates are normalized? In which coordinates are they normalized?
- If the model can generalize to video games, it is worth trying generalizing to CARLA and evaluating it in a closed-loop manner.
- The short name 'DriveGPT4' is not very appropriate, as the core method is finetuned from LLaMA 2 and does not use GPT4.

---

> ### Author Response · Authors · 2023-11-21
> **Response to Reviewer yQKg (Part 1/2)**
>
> We thank the reviewer for detailed comments and suggestions. Our responses to the comments and questions of the reviewer are listed below.
>
> **1. The improvement of the fixed question-answering experiment is limited (82.1->82.4, especially considering this is evaluated by ChatGPT with great uncertainty)**
>
> Evaluation with ChatGPT/GPT4 is commonly conducted in many recent papers about multimodal LLM [1,2], since the generated texts are hard to evaluate considering that they have complicated semantic meanings/logic. Therefore, leveraging ChatGPT for evaluation should be a good way for comparison. In addition, we run the evaluation with ChatGPT for multiple times for more accurate measurement.
>
> **2. LLMs take much longer inference time to output the much longer answers compared to small models and the original QAs. The computational complexity is not that important at the current stage but is still valuable for illustration**
>
> Thanks for the comment. Currently, DriveGPT4 can run at 1~2 FPS on a single V100 GPU card with fp16. The efficiency could be further improved in the futher.
>
> **3. The additional question-answering experiment demonstrates the effectiveness of DriveGPT4 which is finetuned on the generated data. Then how about finetuning ADAPT and other works on the data? The direct comparison seems unfair**
>
> ADAPT only takes video frames as input, so it cannot understand texts as input. It is trained to output two fixed sentences, one for vehicle state description and another for justification. Thus ADAPT cannot be trained with QA pairs, and we cannot finetune ADAPT by our generated dataset. This limits its flexibility for answering other format questions raised by human users. DriveGPT4 shows more advantages for handling this thanks to the multimodal LLM.
>
> **4. For the control prediction task, DriveGPT4 takes historical control signals as input while others do not. It can be observed from the ablation study, that the results of 'No control signals' are close to ADAPT-32. The contribution comes from the history information, rather than LLM, which is unfair.**
>
> Thanks for this good question. In the revised manuscript, history information is removed from the input. Now DriveGPT4 makes prediction only based on video frames and input human questions. Please refer to Table.3 in the revised manuscript for the new results. From the results, we can see that DriveGPT4 still outperforms the previous SOTA ADAPT (8-frames for fair comparison).
>
> **5. It is an open question if flexible answers can help real-world applications of autonomous driving.**
>
> Yes. Flexible questions can be used for better human-user interaction, by providing flexible interface for users. So that users can better understand the vehicle action, and this improves the vehicle transparency and trustworthiness. Besides, flexible QAs can train the model to better understand the driving scenarios, which is helpful for the fixed-QAs as well.
>
> **6. It is not the first work on interpretable end-to-end autonomous driving**
>
> Sorry for the typo. This problem has been fixed in the revised manuscript.
>
> **7. From the very beginning, is interpretability indeed the unsolved problem that hinders the commercialization and development of autonomous driving**
>
> Yes. One critical problem that prevents commercialized autonomous driving is that autonomous vehicles are black boxes. The decisions and actions of autonomous vehicles cannot be understood by humans, which raises severe ethical and law issues. Once the autonomous vehicle leverages deep learning modules, the transparency and trustworthiness of it is always an inevitable problem in the way of commercialized autonomous driving.  LLM could be a potential tool to open this black box by training the LLM to explain for itself when it makes a decision to control the vehicle.
>
> **References**
>
> [1] Liu, Haotian, et al. "Visual instruction tuning." NIPS, 2023.
>
> [2] Luo, Ruipu, et al. "Valley: Video Assistant with Large Language model Enhanced abilitY." arXiv:2306.07207, 2023.

---

> ### Author Response · Authors · 2023-11-21
> **Response to Reviewer yQKg (Part 2/2)**
>
> **8. Why not use ground truth from the BDD dataset? The detection results from YOLOv8 could be inaccurate. I am wondering if there is a truck in the provided figure (Table 1).**
>
> The original BDD-K dataset only has ground truth object bounding boxes for keyframes, which is not enough for our data processing. Thus we need to detect objects in each frame by ourselves. Thanks for pointing out the inaccurately detected truck. We further tuned the settings of YOLO8 and fixed the problem. Inaccurate detection happens sometimes and cannot be completely avoided. However these incorrect detected objects do not seriously affect the QA pairs generated by ChatGPT. The overall quality of the additional QA pairs generated by ChatGPT is assured.
>
> **9. How bounding box coordinates are normalized? In which coordinates are they normalized?**
>
> The bounding boxes (x_min,x_max,x_max,y_max) are normalized based on the image size. This follows LLaVA [1] for instruction tuning data generation.
>
> **10. If the model can generalize to video games, it is worth trying generalizing to CARLA and evaluating it in a closed-loop manner.**
>
> At this stage, DriveGPT4 does not work on close-loop control. The problems of applying LLM for close-loop control, such as drifting, consistent prediction, efficiency, are not well studied  yet. This is a promising future research topic, and we will investigate it in the next step.
>
>
> **11. The short name 'DriveGPT4' is not very appropriate, as the core method is finetuned from LLaMA 2 and does not use GPT4.**
>
> Thanks for the comment. The name “DriveGPT4” does not derive from its utilized LLM. “GPT” here indicates we want to make the autonomous vehicle able to chat with human users in a similar way to ChatGPT/GPT4. Besides, ChatGPT is used for data engineering in our project. Therefore, the model is named DriveGPT4.
>
> **Reference**
>
> [1] Liu, Haotian, et al. "Visual instruction tuning." NIPS, 2023.

---

> > ### Comment · Reviewer_yQKg · 2023-11-22
> > **Response to Authors**
> >
> > Thanks for the rebuttal. I also acknowledge the huge effort to revise papers including figures, writings, and re-running experiments without historical action prompts. But honestly, no highlight of changes makes comparing to the original draft really difficult and time-consuming. After reading other reviewers' comments and the replies, I still have a few comments.
> > - The ''no historical action'' experiments are good for comparison. However, Tab.1 and the details described right above Sec. 4 are not revised accordingly. I assume there should not be control signals in these places. Besides, the third question of Fixed QAs in Tab. 1 is weird (Question: What are the people in the image doing? Answers: historical speed and turning angles).
> > - By comparing ''No ChatGPT QA'' and ''DriveGPT4'' in Tab. 6, results in both the original draft and current version, all show that the Fixed QAs already demonstrate great performance, which means the improvements from more flexible questions are marginal. I think this should be clearly stated in the paper, that the effectiveness of LLM on fixed QA and action prediction is still limited at the current stage.
> > - In terms of the "interpretability hinders the way of commercialized autonomous driving", I sincerely disagree with the statement. I admit that interpretability is an issue when learning modules are applied in planning. But current commercialized autonomous driving usually doesn't involve a lot of learning-based planning, most optimization-based.
> > - I also agree with Reviewer 2Geq that fine-tuning Video-LLaMA should be done for a fair comparison in this work.
> >
> > Concerning the above issues, I summarize the main contribution of this work as, (1) using ChatGPT to expand the QAs upon BDD-X datasets, (2) building a video-language model based on LLM for flexible QA and enhanced human interaction ability. Further contributions or claims need to be very careful to make them sound.

---

> > > ### Author Response · Authors · 2023-11-22
> > > **Response to Reviewer yQKg**
> > >
> > > Thanks for your prompt feedback about our revision. For your questions:
> > >
> > > - Thanks for the information, we have revised the manuscript accordingly. For "historical action", they are not accessed by the LLM, but they are still used by ChatGPT for QA pairs generation. Thus, quoting "control signals" in Tab.1 and the end of section 3 is reasonable.
> > >
> > > - The purpose of using ChatGPT QA pairs is to enable DriveGPT4 to answer human questions in more patterns, instead of in the fixed format like BDD-X QAs. We add more contents in the ablation part (section 5.4), which is quoted here "QA pairs generated by ChatGPT enable DriveGPT4 to answer human questions in more flexible patterns, and slightly enhance the QA ability of BDD-X questions."
> > >
> > > - Thanks for the information. If the planning/control module is conducted by rule-based methods, interpretability is not that important indeed. But considering learning-based methods can handle diverse scenarios with much more complexity, they have the potential for better planning modules. As long as learning-based methods are implemented, interpretability is the key to realize transparent and trustworthy autonomous driving systems.
> > >
> > > - Thanks for the suggestion. We will add it into the comparison in a later stage considering limited time and resources. Valley is proved to be better than Video-LLaMA for general video understanding, thus in the past we directly adapt Valley for our task.

---

### Official Review · Reviewer_u2wu · 2023-10-31

**Soundness:** 3 good
**Presentation:** 4 excellent
**Contribution:** 3 good
**Rating:** 6
**Confidence:** 2

**Summary:**

In the paper, the authors made a contribution by introducing a new image-and-language dataset derived from the BDD-X dataset and enriched using ChatGPT. This customized visual instruction tuning dataset is specifically designed for the application of large language models (LLMs) in autonomous driving. Their system, DriveGPT4, uses this dataset for fine-tuning, serving as a baseline in the field. Notably, DriveGPT4 demonstrates good zero-shot generalization capabilities, akin to the performance metrics observed with ChatGPT. This research offers a new focus on achieving interpretability in end-to-end autonomous driving systems through the use of LLMs. The DriveGPT4 can process multi-modal input data and provide text responses as well as predicted control signals.

The key contributions are 1) a new vision-language dataset for autonomous vehicle; 2) a new chatGPT style system trained on the new dataset; 3) such chatGPT style system DriveGPT4 performs better than alternative baselines on the same dataset, can produce something that ChatGPT variant (GPT-4v) can do. I recognize that GPT-4v was not available at the time when this paper was done.

**Strengths:**

1. The paper breaks new ground by applying large language models (LLMs) like ChatGPT (GPT-4) to the domain of autonomous driving. It offers an invaluable resource in the form of a customized visual instruction tuning dataset, setting a robust baseline for future research that aims to incorporate LLMs into autonomous systems.

2. One of the standout features of the proposed DriveGPT4 system is its impressive zero-shot generalization capabilities. This ability to adapt to previously unseen scenarios mirrors the robustness and flexibility observed in ChatGPT, making it a compelling advance in the field.

**Weaknesses:**

1. A notable limitation of the paper is its focus on a dataset comprised solely of salient images related to autonomous driving. This does not accurately represent real-world conditions, where sensors capture a multitude of irrelevant or non-critical images. This selective approach raises questions about the model's susceptibility to overfitting and its reliance on human oversight for attention guidance. Future work could benefit from evaluating the fine-tuned LLM on a more diverse set of images to assess the model's generalization capabilities beyond the curated dataset.

2. The paper's scope could be considered narrow, as it only tests the fine-tuned LLM on the custom-created BDD-X dataset. Extending evaluations to include other benchmark datasets like Waymo Open could provide a more comprehensive understanding of the model's applicability and robustness in different autonomous driving scenarios. The current evaluation strategy thus limits the paper's contributions to a more confined context.

3. The paper may be perceived as over-ambitious in its claims, particularly given the title "DriveGPT4," which implies a comprehensive solution for Level 4 autonomous vehicles. However, the proposed system focuses narrowly on generating control signals, neglecting other essential aspects like perception, planning, and behavior. This narrow focus limits the model's utility in real-world applications. Additionally, the paper does not sufficiently demonstrate the model's ability to address long-tail or zero-shot cases, further constraining its practical relevance. While DriveGPT4 shows promise for specific tasks like auto-labeling, its current form falls short of making it a broadly applicable solution in the autonomous driving ecosystem.

**Questions:**

Could you elaborate on the measures taken to assess whether the fine-tuned LLM retains its pre-existing knowledge base or if it has overfitted to the custom dataset? Specifically, has the model's performance been evaluated on diverse image sets from alternate data sources to gauge its generalization capabilities?

How does the DriveGPT4 model, trained on the BDD-X dataset, perform on other benchmark datasets like Waymo Open, particularly in tasks such as behavior prediction?

Could you address how the proposed system manages the issue of hallucination? Have there been instances where the model exhibited hallucinatory behavior, and if so, how was this mitigated?

---

> ### Author Response · Authors · 2023-11-21
> **Response to Reviewer u2wu**
>
> We thank the reviewer for detailed comments and suggestions. Our responses to the comments and questions of the reviewer are listed below.
>
> **1. A notable limitation of the paper is its focus on a dataset comprised solely of salient images related to autonomous driving.**
>
> At this stage, this paper is an early research to study the possibility of using LLM for autonomous driving, thus real-world conditions with complicated scenarios are not the focus of this paper. We try to make this model work on the specific dataset first before it can be adapted to more generalized autonomous driving situations.
>
> **2. The paper's scope could be considered narrow, as it only tests the fine-tuned LLM on the custom-created BDD-X dataset.**
>
> Since this paper aims to study interpretable end-to-end autonomous driving, we need labels for control signals as well as semantic explanations in human language. The data source is very limited and BDD-X dataset is the best option for us at this stage. Besides, BDD-X contains 16K video clips for training and over 2K video clips for testing, which contains various weather and driving scenarios. The experiments on this dataset should be sufficient to demonstrate the effectiveness of DriveGPT4. In the next step, we plan to collect/generate/annotate more data and further tune the LLM for better results.
>
> **3. The paper may be perceived as over-ambitious in its claims, particularly given the title "DriveGPT4," which implies a comprehensive solution for Level 4 autonomous vehicles.**
>
> Thanks for the comment. In the original manuscript, there do exist some overclaims, and they have been fixed in the revised manuscript. The name “DriveGPT4” is inspired by Mini-GPT4, where “4” represents multimodality, and it does not indicate Level-4 autonomous driving ability. We have highlighted this delaration in the revised manuscript.
>
> **4. Could you elaborate on the measures taken to assess whether the fine-tuned LLM retains its pre-existing knowledge base or if it has overfitted to the custom dataset? Specifically, has the model's performance been evaluated on diverse image sets from alternate data sources to gauge its generalization capabilities?**
>
> In this paper we aim to study interpretable end-to-end autonomous driving which requires text and control signal labels, and BDD-X is the best option for this purpose. Thus at this stage we can only do comprehensive evaluations on the BDD-X dataset. For generalization ability, this is not the main focus of this paper. DriveGPT4 shows some zero-shot generalization ability, which is visualized in the qualitative results section. Large-scale zero-shot generalization on additional datasets can be an interesting research problem in the next step.
>
> **5. Could you address how the proposed system manages the issue of hallucination? Have there been instances where the model exhibited hallucinatory behavior, and if so, how was this mitigated?**
>
> Hallucinations are noticed during our experiments. Sometimes the model states some objects while the object does not actually exist, e.g., traffic light, pedestrians. Adding additional instruction tuning data generated by ChatGPT relieves the hallucination issue to some extent. We think large amounts of high-quality data are the key for handling hallucinations.

---

> > ### Comment · Reviewer_u2wu · 2023-11-22
> >
> > Thanks for explaining this. This makes sense. I would like to give the benefits of doubts to the authors though I will keep my rating.

---

### Meta-Review · Area_Chair_EW19 · 2023-12-07

**Metareview:**

The paper proposes DriveGPT4 for end-to-end autonomous driving with LLMs in an interpretable manner. The model is trained on a visual instruction tuning dataset (BDD-X enhanced with chatGPT). Three out of four reviewers are critical about the work, where the main criticisms include limited technical contribution and lack of experimentation that support the claims (e.g. interpretability). One reviewer is mildly positive, but they didn't champion for the paper during the discussion. Consequently, my recommendation is to reject the manuscript.

**Justification For Why Not Higher Score:**

Most of the reviewers (3 out of 4) remained unconvinced about the paper.

**Justification For Why Not Lower Score:**

N/A

---

### Decision · Program_Chairs · 2024-01-16

Reject